# Fine-Needle Aspiration Cytology and Histological Types of Thyroid Cancer in the Elderly: Evaluation of 9070 Patients from a Single Referral Centre

**DOI:** 10.3390/cancers13040907

**Published:** 2021-02-22

**Authors:** Anello Marcello Poma, Elisabetta Macerola, Alessio Basolo, Valerio Batini, Teresa Rago, Ferruccio Santini, Liborio Torregrossa

**Affiliations:** 1Department of Surgical, Medical, Molecular Pathology and Clinical Area, University of Pisa, 56126 Pisa, Italy; marcellopoma@gmail.com (A.M.P.); elisabetta.macerola@for.unipi.it (E.M.); 2Department of Clinical and Experimental Medicine, University of Pisa, 56126 Pisa, Italy; alessio.basolo@med.unipi.it (A.B.); rago@endoc.med.unipi.it (T.R.); ferruccio.santini@med.unipi.it (F.S.); 3Section of Laboratory Medicine, University Hospital of Pisa, 56126 Pisa, Italy; v.batini@ao-pisa.toscana.it; 4Section of Pathology, University Hospital of Pisa, Via Roma 57, 56126 Pisa, Italy

**Keywords:** thyroid nodules, thyroid cancer, elderly patients, FNA

## Abstract

**Simple Summary:**

Elderly patients have a high prevalence of thyroid nodules, and their management should consider the presence of comorbidities, which are frequent in this age group. In this retrospective monocentric study, we analyzed data of more than 13,000 nodules in order to highlight differences between the elderly and the general population in terms of cytological and histological diagnoses. Thyroid nodules in the elderly are more often benign than in younger patients. Nevertheless, in case of malignancy, follicular-derived well-differentiated tumors are almost always diagnosed in younger patients. Instead, elderly patients more often have tumors with aggressive histotypes. In addition, even in presence of well-differentiated tumors, elderly patients present a higher rate of high-risk pathological features.

**Abstract:**

Background. The prevalence of thyroid nodules increases with age. Their management takes into account the presence of co-morbidities, which are frequent among the elderly. We sought to highlight the differences between the elderly and the general population in cytological and histological diagnoses. Methods. In this retrospective cohort study, we gathered 13,747 nodule data and compared cytological and histological diagnoses between patients aged over 65 years and a control group. Results. Elderly patients had a higher prevalence of cytologically benign nodules and, consequently, they were less frequently subject to surgery. However, there were no differences in terms of malignancy-risk after surgery. At histology, elderly patients often presented aggressive histology such as medullary thyroid carcinoma, poorly-differentiated and anaplastic cancer, tall cell variant of papillary thyroid carcinoma and Hürthle cell carcinoma. Even in presence of well-differentiated cancer, older patients had higher rates of local invasiveness, lateral lymph node involvement and vascular invasion. Conclusion. Thyroid nodules in elderly patients represent a challenging entity since they are very often benign, but, in case of malignancy, aggressive histotypes and high-risk features are more frequent. Therefore, presurgical characterization of nodules in older patients is crucial and might require strict monitoring.

## 1. Introduction

The prevalence of thyroid nodules in the adult population is growing, with reported ranges between 26 and 67% [1]. Thyroid nodules in the elderly represent a frequent condition, and various epidemiological analyses have demonstrated a positive correlation between thyroid nodule formation and advanced age [2]. This increase is partly due to extended use of ultrasonographic (US) examinations of the head and neck region in routine clinical care, especially in the elderly population [3]. According to the World Health Organization, aging is influenced by many individual and cultural factors. Therefore, the concepts of “old” or “elderly” cannot be considered universal, although the term “elderly” is generally referred to the chronological age population group older than 65 years [4].

Fine-needle aspiration cytology (FNAC) represents the most important diagnostic tool for thyroid nodule evaluation. Several authors have reported that the rate of benign cytology in patients older than 65 years could be higher than usual [5,6]; on the other hand, it is well known that advanced age is one of the factors that negatively influences thyroid cancer prognosis [7]. The surgical rate for cytologically suspicious and malignant nodules in the elderly can be affected by the presence of other co-morbidities that are more frequent than in other age groups. In these elderly patients, a conservative approach might be preferred, whenever possible, to avoid risks and complications associated with surgery [8]. This study describes a large series of thyroid nodules from elderly patients referring to a single institution. Our aim was to compare the distribution of cytological and histological diagnoses between the elderly patients and the general population. In addition, we evaluated how malignancy rates and tumor clinico–pathological characteristics in the elderly can differ from the general population. 

## 2. Results

### 2.1. Cytology

We collected a total of 9858 nodules belonging to 6432 elderly patients and 3889 nodules belonging to 2638 patients of the control group. Considering only the most clinically relevant nodule per patient, lesions in the elderly were classified as follows: 867 TIR 1, 166 TIR 1C, 4204 TIR 2, 842 TIR 3A, 204 TIR 3B, 60 TIR 4 and 89 TIR 5. Instead, in the control group, there were 262 TIR 1, 125 TIR 1C, 1568 TIR 2, 382 TIR 3A, 151 TIR 3B, 55 TIR 4 and 95 TIR 5.

The rate of males in the elderly group was significantly higher (28.6% vs. 21.7%, with a female/male ratio of 2.5 and 3.6 respectively, *p* < 0.001), as well as the occurrence of multiple aspirated nodules (38.4% vs. 33.3%, *p* < 0.001). The nodules in the elderly were larger (26.9 mm ± 13.9 vs. 24.4 mm ± 12.4, *p* < 0.001) and more frequently non-diagnostic (i.e., TIR 1, 13.5% vs. 9.9%, *p* < 0.001). The rate of non-diagnostic cystic nodules (TIR 1C) was instead higher in the control group (4.7% vs. 2.6%, *p* < 0.001).

As regards the diagnostic categories, the nodules in the elderly were more frequently benign (TIR 2), whereas the proportion of nodules diagnosed as high-risk indeterminate (TIR 3B), suspicious (TIR 4) or malignant (TIR 5) was higher in the control group. The details are reported in Figure 1.

### 2.2. Molecular Status

The molecular status of 69 (1.1%) elderly patients and 58 (2.2%) controls were available. Overall, 11 (15.9%) and 9 (15.5%) cases were mutated in the elderly and in the control groups respectively. As expected, the mutation rate increased according to the diagnostic category, with no differences between the elderly and the control group. Only *RAS*-like mutations were found in the TIR 2, TIR 3A and TIR 3B categories, while *BRAF* V600E mutation was detected in TIR 4 and TIR 5 lesions. The details are reported in Table 1.

### 2.3. Histopathological Evaluation

Overall, 514 and 543 histological diagnoses were available for the elderly and controls respectively (Table 2). Consequently, the surgical rate was higher in the control group (20.6% vs. 8.0%, *p* < 0.001). Considering each cytological diagnostic category, the surgical rate was significantly higher in the controls than in the elderly in TIR 1/1C, TIR 2, TIR 3A, TIR 3B (Figure 2), while no differences were observed in TIR 4 and TIR 5. In particular, the greatest differences were in TIR 3A and TIR 2, with respectively a 3-fold and a 2-fold surgical increase in the younger patients.

The type of surgery was more conservative in the control group with a total thyroidectomy rate of 88.2% versus 92.3% (*p* = 0.03). The global malignancy rate was 42.2% with no differences between the two groups. Furthermore, there were no differences between the elderly and the controls in the malignancy rate within different cytological diagnostic categories (Figure 3).

As regards malignant tumours, follicular-derived thyroid cancer (FDTC) was more common in the control group (97.1% vs. 89.0%, *p* = 0.002); instead, medullary thyroid cancer (MTC) was more often diagnosed in elderly patients (8.6% vs. 2.1%, *p* < 0.001). Among FDTC, poorly-differentiated and anaplastic carcinoma were more frequent in elderly patients (5.9% vs. 1.3%, *p* = 0.02) (Figure 4).

Hürthle cell carcinoma (HCC) and tall cell variant of PTC (TCPTC) were frequently observed in the elderly, whereas the classic variant of PTC (CVPTC) was more common in the control group (Figure 5 and Figure 6).

The features of PTC and follicular carcinoma (including HCC) were compared between elderly and control patients. PTC in the elderly were larger, less frequently encapsulated and, whenever present, lymph node metastases commonly involved lateral lymph nodes. The details are reported in Table 3.

As regards follicular carcinoma, minimally invasive tumors without vascular invasion were common in the control group, whereas the proportion of angioinvasive lesions was higher in the elderly. Finally, widely invasive tumors were observed in eld erly patients only (Table 4).

## 3. Discussion

Thyroid nodules are very frequent, being present in about one-third of the general population [9]. Numerous evidences suggest that the prevalence of thyroid nodules and the number of nodules per patient increase with age [9,10]. This higher prevalence is due at least in part to a more frequent use of imaging methods (i.e., ultrasound, computed tomography, magnetic resonance) in elderly patients [11,12].

In order to assess the distribution of the cytological categories and their relation with histological outcomes in elderly patients, we evaluated cytological and histological diagnoses in 6432 patients over 65 years of age compared to a control group of 2638 patients younger than 65 years.

The cytological diagnoses of more than 13,000 nodules were collected. As previously reported [5,13], the rate of indeterminate, suspicious, or cytologically malignant nodules were lower in elderly patients. On the contrary, benign cytology was very common in the elderly. The different rate of benign cytology was not observed by Rossi and colleagues, but, in this case, elderly was defined as patients aged 70 or more years; in addition, pediatric patients were omitted from the general population [5]. These factors could account for the different results.

Non-diagnostic nodules were more prevalent in the elderly, but non-diagnostic cystic nodules were frequent in the control group. As expected, the surgical rate of these nodules was very low in both groups. However, non-diagnostic nodules submitted to surgery proved to be very often malignant, suggesting hyper-selection due to US or clinically suspicious features.

The surgical rate was almost three times higher in the control than in the elderly group in accordance with the higher rate of suspicious or malignant cytological diagnoses. However, this difference cannot entirely account for the higher rate of malignant cytology in younger patients. Indeed, we observed that whenever a malignancy is detected at cytology, the surgical rate is comparable; on the other hand, in presence of benign cytology or low-risk lesions, surgery in the elderly is often avoided. The surgical approach was more conservative in younger patients.

At histology, the malignancy rate was not different in the two groups, ranging from about 10% in TIR 2 to 100% in TIR 5, with an overall 42.2% of nodules that proved to be malignant. The malignancy rate in benign and indeterminate categories is higher than expected, suggesting again the accurate clinical selection of patients submitted to surgery. In younger patients, however, more than 97% of carcinomas were follicular-derived, especially WDTC, with PTC covering almost 87% of all malignancies. On the contrary, the types of cancer observed in the elderly were diversified. At a glance, the rate of non-follicular-derived malignancies was considerable, with MTC representing 8.6% of cases. ATC was present only among the elderly, consistently with the fact that the mean age at diagnosis of ATC is 65 years [14]. PDTC and ATC considered as a whole were observed in almost 6% of FDTC in the elderly compared to 1.3% in the control group. As previously reported, HCC diagnosis was more frequent among the elderly [15]. It is worth noting that older patients also showed a trend for a lower PTC rate. Finally, among the PTC variants, CVPTC was the predominant subtype in the control group, while FVPTC was the most common variant in the elderly. The significant decrease of CVPTC in elderly patients could be explained by the rise of TCPTC, which are typically associated with older age [16].

Another interesting point is that, for a given histological subtype, features of local aggressiveness were more often observed in the elderly. These findings hold true for both PTC and follicular patterned cancers other than FVPTC (i.e., FTC and HCC). Accordingly, we observed PTC with greater size and less commonly encapsulated. The findings on the involvement of lymph nodes are quite controversial; in fact, elderly patients with PTC are frequently N0, but, when metastases are present, lateral lymph nodes are often involved. Similarly, FTC and HCC were never widely invasive in the control group, with the great majority being minimally invasive without angioinvasion. On the other hand, angioinvasive forms were frequently detected in older patients, and a non-negligible number of cases showed massive loco-regional invasion.

Our study suffers from some limitations. First, the complete clinical records of pre-cytological examinations were not available; consequently, we might have underestimated the cases of multi-nodularity. Nevertheless, we referred to aspirated nodules, which account for clinically suspicious nodules and should not be unbalanced due to different ages. The major strength of the study is the sample size, which is the largest ever reported regarding this specific topic.

In summary, the higher prevalence of thyroid nodules among the elderly, more often submitted to imaging techniques, could explain the higher rate of benign cytology. However, whenever a malignancy is detected, the histological types are more diversified, and aggressive features are typically present even in well-differentiated carcinomas.

## 4. Materials and Methods

### 4.1. Patient Cohorts

All patients older than 65 years who underwent FNA cytology at the Unit of Endocrinology of the University Hospital of Pisa between May 2015 and September 2020 were included in the study. The patients aged 65 or younger submitted to FNA cytology between May and December 2015 were used as the control group. The study conforms to the Principles of the Helsinki Declaration of 1975 and its subsequent updates. Both informed and surgical consent was achieved one day before the operation.

Cytology specimens were stained with Papanicolaou and evaluated by expert thyroid cytopathologists.

Lesions were classified according to the Italian consensus for the classification and reporting of thyroid cytology, which includes six diagnostic categories, each of them corresponding to a different risk of malignancy. The above-mentioned cytological categories are defined as follows: TIR 1/TIR 1C—non-diagnostic/non-diagnostic cystic; TIR 2—non-malignant/ benign; TIR 3A—low-risk indeterminate lesion; TIR 3B—high-risk indeterminate lesion; TIR 4—suspicious for malignancy; TIR 5—malignant [17]. In the case of multiple aspirated nodules per patient, the most clinically relevant diagnosis was considered. In detail, the nodule with the highest cytological suspicion for malignancy was considered; in the case of nodules with the same cytological diagnosis, the greater in size was taken into account.

For a subset of patients, molecular analysis was performed upon the endocrinologist’s request. The mutational status of hot spot regions in the *BRAF* and *RAS* genes was evaluated on DNA samples purified from cytological slides using allele-specific real-time PCR (EasyThyroid, Diatech Pharmacogenetics, Jesi, Italy).

A subset of patients underwent surgery, and the histological diagnosis according to the 4th edition of the World Health Organization classification of endocrine tumors [18] was available.

### 4.2. Statistical Analyses

Continuous variables were analyzed by the Student’s t-test. Pearson’s chi-squared test with analysis of standardized residuals was used for categorical variables. Fisher’s exact test was run whenever appropriate. *p*-values below 0.05 were considered significant. All analyses were performed in the R environment (version 4.0.2, https://www.r-project.org/, last accessed December 2020).

## 5. Conclusions

The surgical and clinical management of thyroid nodules in older patients might be particularly challenging. Surgery in older patients is often avoided on account of the presence of other co-morbidities, making it necessary to carefully evaluate and monitor any suspicious nodules. Since in presence of malignancy, elderly patients have often aggressive tumors, it could be worth integrating cytological findings with US features and possibly molecular markers to better define the pre-surgical risk stratification of patients.

## Figures and Tables

**Figure 1 cancers-13-00907-f001:**
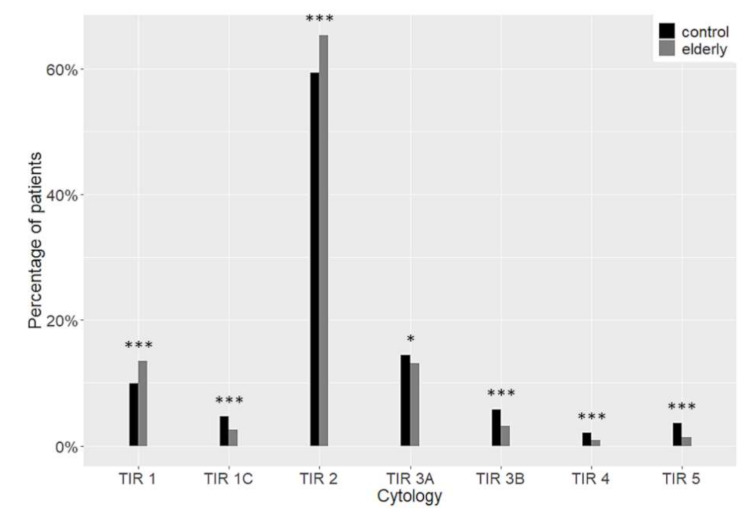
Distribution of cytological diagnoses. Differences between the elderly (grey) and the control group (black) are highlighted. * *p* < 0.05 *** *p* < 0.001.

**Figure 2 cancers-13-00907-f002:**
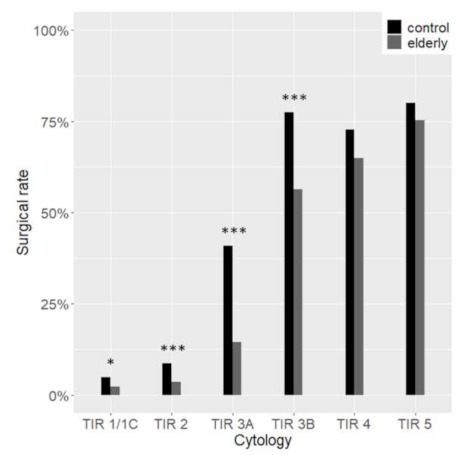
Surgical rate according to cytological categories. In grey are elderly patients; in black are the control group. * *p* < 0.05 *** *p* < 0.001.

**Figure 3 cancers-13-00907-f003:**
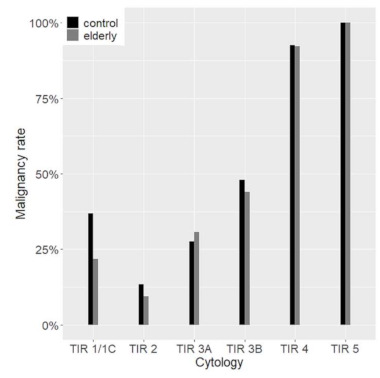
Malignancy rate according to cytological categories. In grey and black are the elderly and the control group respectively.

**Figure 4 cancers-13-00907-f004:**
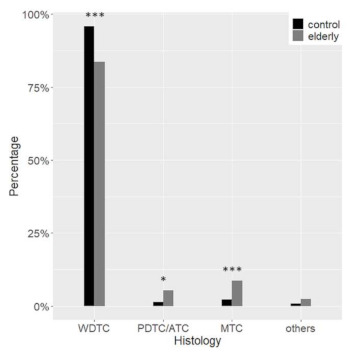
Histological diagnoses. Differences between the elderly (grey) and the control (black) group are highlighted. WDTC, well-differentiated thyroid carcinoma; PDTC, poorly-differentiated thyroid carcinoma; ATC, anaplastic thyroid carcinoma; MTC, medullary thyroid carcinoma * *p* < 0.05 *** *p* < 0.001.

**Figure 5 cancers-13-00907-f005:**
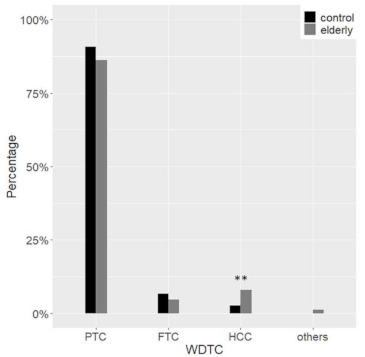
Follicular-derived well-differentiated tumors: differences between the elderly (grey) and the control (black) group. WDTC, well-differentiated thyroid carcinoma; PTC, papillary thyroid carcinoma; FTC, follicular thyroid carcinoma; HCC, Hürthle cell carcinoma ** *p* < 0.01.

**Figure 6 cancers-13-00907-f006:**
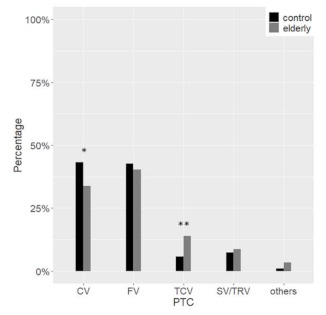
Differences in papillary thyroid carcinoma variants between the elderly (grey) and the control (black) group. PTC, papillary thyroid carcinoma; CV, classic variant; FV, follicular variant; TCV, tall cell variant; SV, solid variant; TRV, trabecular variant * *p* < 0.05 ** *p* < 0.01.

**Table 1 cancers-13-00907-t001:** Mutational status of elderly and control patients according to cytological diagnostic categories.

Cytology	Elderly (*n*. 69)			Control (*n*. 58)		
	*n*. wild-type (%)	*n*. mutated (%)	mutation	*n*. wild-type (%)	*n*. mutated (%)	mutation
TIR 1/1C	2/2 (100%)	-		3/3 (100%)	-	
TIR 2	12/13 (92.3%)	1/13 (7.7%)	*KRAS* cod 61	13/14 (92.9%)	1/14 (7.1%)	*NRAS* cod 61
TIR 3A	32/35 (91.4%)	3/35 (8.6%)	all *NRAS* cod 61	22/25 (88%)	3/25 (12%)	*BRAF* p.K601E *KRAS* cod 61 *NRAS* cod 61
TIR 3B	11/15 (73.3%)	4/15 (26.7%)	*HRAS* cod 61 *KRAS* cod 61 2 *NRAS* cod 61	9/11 (81.8%)	2/11 (18.2%)	all *HRAS* cod 61
TIR 4	1/3 (33.3%)	2/3 (66.7%)	all *BRAF* p.V600E	2/2 (100%)	-	
TIR 5	-	1/1 (100%)	*BRAF* p.V600E	-	3/3 (100%)	all *BRAF* p.V600E
Total	58 (84.1%)	11 (15.9%)		49 (84.5%)	9 (15.5%)	

**Table 2 cancers-13-00907-t002:** Summary of histological diagnoses. Raw data and percentages for both elderly and the control group are reported.

Histology				Number of Cases (%)	
Cell of Origin	FDTC Differentiation	WDTC Type	PTC Variant	Elderly (*n*. 209)	Control (*n*. 237)
			CVPTC	51 (33.8%)	89 (43.2%)
			FVPTC	61 (40.4%)	88 (42.7%)
			TCVPTC	21 (13.9%)	12 (5.8%)
			SV/TRVPTC	13 (8.6%)	15 (7.3%)
			others	5 (3.3%)	2 (1.0%)
		PTC		151 (86.3%)	206 (90.8%)
		FTC		8 (4.6%)	15 (6.6%)
		HCC		14 (8.0%)	6 (2.6%)
		others		2 (1.1%)	0
	WDTC			175 (94.1%)	227 (98.7%)
	PDTC			9 (4.8%)	3 (1.3%)
	ATC			2 (1.1%)	0
FDTC				186 (89.0%)	230 (97.1%)
MTC				18 (8.6%)	5 (2.1%)
others				5 (2.4%)	2 (0.8%)

FDTC, follicular-derived thyroid carcinoma; WDTC, well-differentiated thyroid carcinoma; PTC, papillary thyroid carcinoma; CVPTC, classic variant papillary thyroid carcinoma; FVPTC, follicular variant papillary thyroid carcinoma; TCVPTC, tall cell variant papillary thyroid carcinoma; SV/TRVPTC, solid variant/trabecular variant papillary thyroid carcinoma; FTC, follicular thyroid carcinoma; HCC, Hürthle cell carcinoma; PDTC, poorly-differentiated thyroid carcinoma; ATC, anaplastic thyroid carcinoma; MTC, medullary thyroid carcinoma.

**Table 3 cancers-13-00907-t003:** Papillary thyroid carcinoma. Differences in pathological features between elderly and control patients.

	Elderly (*n*. 151)	Control (*n*. 206)	*p*-Value
Size (mean ± sd, mm)	25 ± 16	21 ± 12	<0.001
Gender			0.06
Male	57 (37.7%)	57 (27.7%)	
Female	94 (62.3%)	149 (72.3%)	
Local invasion			0.02
Encapsulated	22 (14.6%)	45 (21.8%)	
Invasive	48 (31.8%)	41 (19.9%)	
Infiltrative	81 (53.6%)	120 (58.2%)	
Pathologic lymph nodes			0.009
N0	19 (44.2%)	12 (22.2%)	
N1a	9 (20.9%)	27 (50.0%)	
N1b	15 (34.9%)	15 (27.8%)	
Nx *	108	152	
Vascular emboli			0.09
yes	41 (27.2%)	39 (18.9%)	
no	110 (72.8%)	167 (81.1%)	
Thyroiditis			0.32
yes	36 (23.8%)	60 (29.1%)	
no	115 (76.2%)	146 (70.9%)	

* Nx cases were not considered for statistics.

**Table 4 cancers-13-00907-t004:** Follicular and Hürthle cell carcinoma. Differences in pathological features between follicular and oncocytic tumors in elderly and control patients.

	Elderly (*n*. 22)	Control (*n*. 21)	*p*-Value
Size (mean ± sd, mm)	47 ± 21	25 ± 9	<0.001
Gender			0.92
Male	6 (27.3%)	7 (33.3%)	
Female	16 (72.7%)	14 (66.7%)	
Histology			0.05
FTC	8 (36.4%)	15 (71.4%)	
HCC	14 (63.6%)	6 (28.6%)	
Local invasion			0.002
MI	8 (36.4%)	18 (85.7%)	
Angioinvasive	10 (45.4%)	3 (14.3%)	
WI	4 (18.2%)	-	

FTC, follicular thyroid carcinoma; HCC, Hürthle cell carcinoma; MI, minimally invasive tumor without angio invasion; WI, widely invasive tumor.

## Data Availability

The data presented in this study are available in the manuscript.

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
