# Peer review of "Fine-Needle Aspiration Cytology and Histological Types of Thyroid Cancer in the Elderly: Evaluation of 9070 Patients from a Single Referral Centre"

_cancers, 2021, doi:10.3390/cancers13040907_

Round 1

Reviewer 1 Report

Fine-needle aspiration cytology and histological types of thyroid cancer in the elderly: evaluation of 9070 patients from a single Referral Centre by
Anello Marcello Poma et al. presents the distribution of thyroid nodules in elderly patients and the difference between the malignant and the general population's pathological characteristics. The present manuscript is the continuation of several previously published research articles. This manuscript's aim vaguely described, and the paper is challenging to follow, poorly and hastily written. The terminologies are not defined clearly. The scoring system needs a clear explanation and defines the TIR. The statistical analysis requires a clear answer as to why student t-test used, why all the data are presented as a percentage and not presenting the raw data and perform statistical analysis on the raw data. The discussion section from lines 150 to 153 requires a clear explanation describing the contrasting results observed between the elderly and younger patients. The methods section requires a detailed description of the methodologies used. The authors have minimally brushed on the surface of the shortfall in the manuscript and suggested an integrated approach using Ultrasound, cytological finding and malignant markers to help to identify high-risk malignancy. Describe the strength and shortcoming of the study. Overall, an article that could be of value in the management of thyroid patients treatments.

Author Response

Fine-needle aspiration cytology and histological types of thyroid cancer in the elderly: evaluation of 9070 patients from a single Referral Centre by Anello Marcello Poma et al. presents the distribution of thyroid nodules in elderly patients and the difference between the malignant and the general population's pathological characteristics. The present manuscript is the continuation of several previously published research articles. This manuscript's aim vaguely described, and the paper is challenging to follow, poorly and hastily written. The terminologies are not defined clearly.

A: The manuscript was revised, and some parts were re-written, including the aim (lines 58-62, 66-72, 102-108, 151-152, 166-172, 174-177, 187-188, 205-209, 2013-219, 234-242, 260-264).

The scoring system needs a clear explanation and defines the TIR.

A: An explanation of the cytological categories was added in the Methods section (lines 234-238).

The statistical analysis requires a clear answer as to why student t-test used, why all the data are presented as a percentage and not presenting the raw data and perform statistical analysis on the raw data.

A: The student’s t-test was used for continuous variables (i.e. patients’ age and size of nodules). As stated in the manuscript, Pearson’s chi-squared test was used for categorical variables, therefore the analyses were performed on raw data. Since we had multiple contingency tables, we also analysed the standardized residuals to evaluate which specific values differed significantly from the expected ones. We also explained that Fisher’s exact text was used whenever appropriate (i.e. when the expected values within a cell of the contingency table were lower than five). As suggested, we also reported raw data (lines 66-70 and Table 2). Should you need any further explanations about the statistical analyses, please do not hesitate to ask.

The discussion section from lines 150 to 153 requires a clear explanation describing the contrasting results observed between the elderly and younger patients.

A: This part was re-written and the results were discussed in the light of other studies (lines 166-172).

The methods section requires a detailed description of the methodologies used.

A: A clear explanation of nodules considered in cases of multi-nodularity was added (lines 239-242).

The authors have minimally brushed on the surface of the shortfall in the manuscript and suggested an integrated approach using Ultrasound, cytological finding and malignant markers to help to identify high-risk malignancy.

A: We agree with the reviewer but we do not have the complete clinical framework. However, in the conclusions we suggest that, since aggressive tumours are very often observed in the elderly, an integrated approach might be particularly beneficial in the pre-surgical management of this subgroup of patients. We have re-phrased the sentence in order to better explain our idea and to avoid speculation (lines 260-264).

Describe the strength and shortcoming of the study. Overall, an article that could be of value in the management of thyroid patients treatments.

A: A paragraph discussing the strengths and limitations of the study was added in the discussion (lines 213-218).

Reviewer 2 Report

The authors have performed a retrospective analysis of the cytological and histological diagnoses and compared them between the elderly and the control group. The findings of the study are not entirely novel and have been reported before. Also, the study data hasn't been presented well and needs to be improved. The strength of the study is the population size in comparison to the similar published studies. 

Comments to the authors:

The authors indicate the data in terms of percentage at maximum places. They should also include the number of patients included in each analysis. Please include this information everywhere as the sample size in population studies is a critical determinant in deciding the data's overall impact. 

Please indicate the number of male and female patients in each group and whether the male/female ratio was significantly different.

How many patients had multiple nodules and metastasis in each group?

The discussion section does not discuss similar studies appropriately. The authors should compare their studies' key observations with other published studies and discuss any differences in their observations and the probable reason for that.

Also, please include a paragraph discussing your study's strength compared to the published studies and the limitations of your study.

Author Response

The authors have performed a retrospective analysis of the cytological and histological diagnoses and compared them between the elderly and the control group. The findings of the study are not entirely novel and have been reported before. Also, the study data hasn't been presented well and needs to be improved. The strength of the study is the population size in comparison to the similar published studies.

Comments to the authors:

The authors indicate the data in terms of percentage at maximum places. They should also include the number of patients included in each analysis. Please include this information everywhere as the sample size in population studies is a critical determinant in deciding the data's overall impact.

A: We added raw data for both cytological and histological diagnoses (lines 66-70 and Table 2 respectively).

Please indicate the number of male and female patients in each group and whether the male/female ratio was significantly different.

A: The female/male ratio was added (lines 71-72), and the specification of the gender was also added in Tables 3 and 4.

How many patients had multiple nodules and metastasis in each group?

A: The data about multiple aspirated nodules had already been reported in the results (lines 72-73). Unfortunately, we do not have the complete records of non-aspirated nodules. In addition, we reported data about lymph node metastases in PTC, but a complete assessment of distant metastases was not available.

The discussion section does not discuss similar studies appropriately. The authors should compare their studies' key observations with other published studies and discuss any differences in their observations and the probable reason for that.

A: The results were discussed in the light of other studies (lines 166-172).

Also, please include a paragraph discussing your study's strength compared to the published studies and the limitations of your study.

A: A paragraph discussing the strengths and limitations of the study was added in the discussion (lines 213-218).

Reviewer 3 Report

This is a single institution review of over 13000 thyroid nodules in elderly and younger (control) groups showing that benign histology is more common among elderly but when malignant tumors occur, they are more aggressive in histologic subtype, and extent of disease. There were very few subjects with molecular testing available, but for the few, there was no significant difference in mutations between elderly and younger subjects. The proportion of anaplastic, poorly differentiated, and tall cell variants was greater among the elderly and there was a significantly greater amount of MTC in the elderly. These observations may explain the greater tumor invasiveness among the elderly. The paper is well written and the data support the conclusions.  

Author Response

This is a single institution review of over 13000 thyroid nodules in elderly and younger (control) groups showing that benign histology is more common among elderly but when malignant tumors occur, they are more aggressive in histologic subtype, and extent of disease. There were very few subjects with molecular testing available, but for the few, there was no significant difference in mutations between elderly and younger subjects. The proportion of anaplastic, poorly differentiated, and tall cell variants was greater among the elderly and there was a significantly greater amount of MTC in the elderly. These observations may explain the greater tumor invasiveness among the elderly. The paper is well written and the data support the conclusions.

A: Thank you.

Round 2

Reviewer 2 Report

The changes incorporated by the authors have improved the manuscript.